# Task-Adapter: Task-specific Adaptation of Image Models for Few-shot Action Recognition

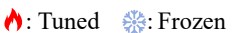

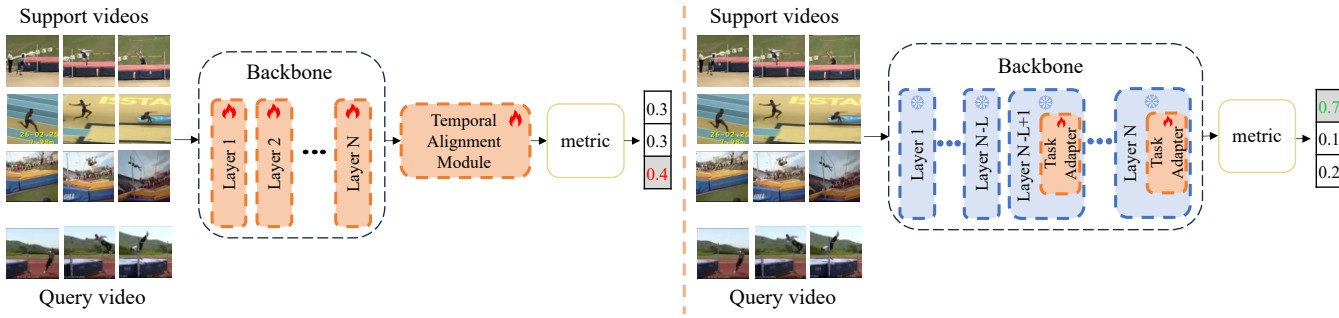

(a) Paradigm of Traditional Few-shot Action Recognition

(b) Paradigm of Our Proposed Task-Adapter

Figure 1: Paradigm comparison between classic few-shot action recognition methods and ours. (a) Existing methods fully fine-tune the feature extractor and combine it with temporal alignment module at feature level. (b) We advance the alignment module into the process of feature extraction particularly considering the internal information of the entire task by introducing tunable task-specific adapters into the frozen pre-trained model.

## ABSTRACT

Existing works in few-shot action recognition mostly fine-tune a pre-trained image model and design sophisticated temporal alignment modules at feature level. However, simply fully fine-tuning the pre-trained model could cause overfitting due to the scarcity of video samples. Additionally, we argue that the exploration of task-specific information is insufficient when relying solely on well extracted abstract features. In this work, we propose a simple but effective task-specific adaptation method (Task-Adapter) for few-shot action recognition. By introducing the proposed Task-Adapter into the last several layers of the backbone and keeping the parameters of the original pre-trained model frozen, we mitigate the overfitting problem caused by full fine-tuning and advance the task-specific mechanism into the process of feature extraction. In each Task-Adapter, we reuse the frozen self-attention layer to perform task-specific self-attention across different videos within the given task to capture both distinctive information among classes and shared information within classes, which facilitates task-specific adaptation and enhances subsequent metric measurement between the query feature and support prototypes. Experimental results consistently demonstrate the effectiveness of our proposed Task-Adapter on four standard few-shot action recognition datasets. Especially on temporal challenging SSv2 dataset, our method outperforms the state-of-the-art methods by a large margin.

## CCS CONCEPTS

• **Computing methodologies → Activity recognition and understanding**;

## KEYWORDS

few-shot action recognition, parameter-efficient fine-tuning, task-specific adaptation

## 1 INTRODUCTION

In recent years, large-scale pre-trained models [19, 32] have demonstrated remarkable performance across various downstream tasks. However, manually collecting hundreds of millions of labeled data and training such large models from scratch for each specific task is impractical. Hence, researchers are increasingly focusing on fine-tuning large pre-trained models on specific tasks [10, 48, 54, 59] to harness their generalization capability. In video understanding, the utilization of large-scale pre-trained models has significantly enhanced the performance of action recognition [15, 21, 28, 29, 40, 47, 49, 54]. Yet, these works typically fine-tune pre-trained models on well-labeled video datasets [5, 11], where the testing and training categories are the same. In realistic scenarios, collecting sufficient and well-labeled video samples is inherently hard. Most problems

manifest in a few-shot manner: there are very few annotated samples for new action categories. Few-shot learning [3, 6, 30, 37, 55] aims to achieve satisfactory classification performance on any few-shot manner unseen class data by training only on seen class data. Therefore, how to leverage the prior knowledge of pre-trained models to improve few-shot action recognition is an imperative challenge.

In few-shot action recognition, a common practice is following the metric based meta-learning strategy [4, 34, 39], which computes classification probability according to the measured distance between query video feature and support prototypes. Consequently, previous works [4, 14, 20, 26, 31, 42, 44, 51] design various temporal alignment modules at the feature level to better match the features of different videos. These works commonly use a pre-trained image model [7, 32] and then fine-tune on the video dataset. An illustration of this traditional paradigm is depicted in Figure 1 (a). However, this traditional paradigm has two drawbacks: First, the majority of works adopt full fine-tuning strategy, i.e., fine-tuning all parameters of the pre-trained model on seen classes and expecting to learn a robust feature extractor which maintains good generalization performance on unseen classes. However, this classic full fine-tuning strategy could overly destroy the prior knowledge of the pre-trained model, leading to catastrophic forgetting and overfitting to seen class data. The second drawback is that most existing works neglect the importance of extracting the task-specific information (i.e., inter-class distinctive information among classes and intra-class shared information within classes). Solely designing temporal alignment modules at highly abstracted feature level could not discover the most discriminative feature for a given few-shot learning task.

To address the first drawback, we turn to the recently emerged Parameter-Efficient Fine-Tuning (PEFT) approach. Originating from the field of NLP [12, 13, 17], PEFT aims to adapt large pre-trained models to downstream applications without fine-tuning all the parameters of the model, while attaining performance comparable to that of full fine-tuning. In the field of computer vision, a series of adapter-tuning works [29, 54] have demonstrated the effectiveness of PEFT for video understanding. Inspired by these works, we argue such methods, which only fine-tune a small number of additional adapter parameters, are particularly well-suited for few-shot action recognition, since they enable better trade-offs between leveraging the generalizable prior knowledge of pre-trained models and incorporating domain-specific knowledge from downstream tasks. However, existing adapter methods mainly focus on temporal modeling and lack the consideration of enhancing task-specific features for the few-shot learning tasks, consequently not suitable to be directly introduced to the few-shot action recognition field. In this work, we propose a novel adapter-based method which further brings the capability of task-specific adaptation to the pre-trained model by reusing the frozen self-attention block to perform the task-specific self-attention across different videos within the given task to capture the most discriminative task-relevant information.

Task-specific adaptation of features has always been a key issue in metric based few-shot learning [18, 22, 44, 55]. A common consensus is that the most discriminative features vary for different target tasks. For example, in distinguishing between "High Jump" and "Long Jump", the body movements performed by the actor is more crucial, whereas in distinguishing between "High Jump" and "Pole Vault", it is more significant whether the actor is holding a long pole. In few-shot action recognition, most previous works [4, 14, 20, 26, 31, 42, 44, 51] focus on the temporal alignment between different video features, but only a few [44] consider task-specific adaptation. Meanwhile, existing works solely apply task-specific methods at the level of well-extracted features. Different from previous works, we propose to perform task-specific adaptation during the process of feature extraction as shown in Figure 1 (b). Specifically, for a Transformer backbone, e.g. ViT, in addition to the original spatial self-attention over the image tokens within the same frame, we further apply the frozen self-attention layer over the image tokens across different frames and different videos to perform both temporal attention and task attention across different videos within a given task. Experimental results reveal that performing task-specific adaptation during the feature extraction could better capture the discriminative task-specific feature compared to only adapting the final features. In summary, our contributions can be outlined as follows:

- We introduce a novel parameter-efficient adaptation method for few-shot action recognition called Task-Adapter, effectively alleviating the issues of catastrophic forgetting and overfitting induced by full fine-tuning.
- To the best of our knowledge, we are the first to propose performing task-specific adaptation during the process of feature extraction, which significantly enhances the discriminative features specific to the given few-shot learning task.
- Extensive experiments demonstrate the superiority and good generalization ability of our Task-Adapter, which achieves new state-of-the-art results on four standard few-shot action recognition benchmarks.

## 2 RELATED WORK

### 2.1 Few-shot Learning

Few-shot learning aims to learn a model that can quickly adapt to unseen classes using only a few labeled data. Existing works can be divided into three categories: augmentation-based [25, 45], metric-based [1, 3, 18, 33, 34, 39, 50, 55, 55] and optimization-based methods [8, 53]. Augmentation-based methods use auxiliary data to enhance the feature representations or generate extra samples [25] to increase the diversity of the support set. Metric-based methods focus on learning a generalizable feature extractor and classify the query instance by measuring the feature distance. Optimization-based methods aim to learn a good network initialization which can fast adapt to unseen classes with minimal optimization steps. Among these approaches, metric-based methods are commonly adopted thanks to its simplicity and superior performance. Additionally, task-specific learning has been explored in image-domain metric-based works [1, 18, 33, 55] to achieve better few-shot learning performance. CTM [18] proposes category traversal module to obtain the most discriminative features within a given C-way K-shot few-shot learning task. FEAT [55] introduces set-to-set function to obtain task-specific visual features for the few-shot learning tasks. However, these methods all introduce task-specific modules at the level of well-extracted image features, which is insufficient for more complex video features due to the additional temporal dimension. In this work, we follow the metric-based paradigm and perform

task-specific adaptation during the feature extraction process to achieve more comprehensive adaptation to video data.

## 2.2 Few-shot action recognition

In few-shot action recognition, most of the previous works[2, 4, 9, 14, 20, 24, 26, 27, 31, 36, 43, 44, 46, 52, 57, 62, 63] fall into the metric-based methods. To solve the problem of temporal misalignment when comparing different videos, a wide series of temporal alignment modules are proposed. OTAM [4] improves the dynamic time warping algorithm to preserve the temporal ordering inside the video feature similarities. TRX [31] proposes to take the ordered tuples of frames as the least matching unit instead of single frames. CMOT [24] uses the optimal transport algorithm to find the best matching between two video features. HyRSM [44] treats the sequence matching problem as a set matching problem for better alignment. Except for the aforementioned works, there are also studies focusing on spatial relations [58] and multi-modal fusion [9, 41, 46]. However, there is still a lack of research on task-specific learning in few-shot action recognition. In this work, we emphasize the vital importance of task-specific adaptation in identifying the most discriminative spatial-temporal features within the provided video samples.

## 2.3 Parameter-Efficient Fine-Tuning

Parameter-Efficient Fine-Tuning (PEFT) is derived from NLP [12, 13, 17], aiming to fine-tune large pre-trained models on downstream tasks in an efficient way (i.e., only fine-tune a small number of parameters while freezing most parameters of the pretrained model), while achieving comparable performance to full fine-tuning. Recently, this method has been introduced to computer vision fields. CLIP-Adapter [10] adds a learnable bottleneck layer behind the frozen CLIP [32] backbone to learn new features for downstream tasks. ST-Adapter [29] proposes a 3D convolution based adapter to model the temporal information. AIM [54] reuses the frozen pre-trained attention layer and applies it to the temporal dimension of the input video. Vita-CLIP [47] introduces learnable prompt tokens into the frozen model to enrich the representation capability. However, the aforementioned PEFT methods do not consider the characteristic of the few-shot learning task, where task-specific adaptation plays a crucial role in distinguishing the query sample from irrelevant categories. In this work, we propose a novel PEFT method to maximize the generalization ability of the pre-trained model and equip the pre-trained image model with the capability of enhancing task-specific features for few-shot action recognition.

## 3 METHODOLOGY

In this section, we first describe how AIM [54] utilizes the pretrained image model ViT to perform action recognition in Section 3.1. Then, in Section 3.2, we extend this method to few-shot action recognition by introducing task-specific adaptation capability. The overview of our method is shown in Figure 3.

## 3.1 Adapting ViT to Action Recognition

Given an RGB video sample $v \in R^{T \times H \times W \times 3}$, a vanilla ViT takes the temporal dimension $T$ as the batch dimension $B$, i.e., there is no exchange of information between frames. Specifically, ViT divides

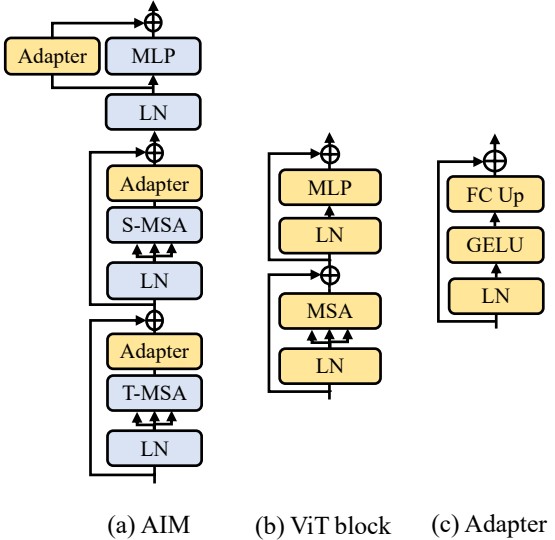

(a) AIM      (b) ViT block     (c) Adapter

**Figure 2: AIM (a) adapts the standard ViT bock (b) by freezing the original pre-trained model (outlined with a blue background) and adding tunable Adapters (c) individually for temporal adaptation, spatial adaptation and joint adaptation.**

each frame into $N = HW/P^2$ patches, where $P$ is the patch size, and transforms the patches into $D$ dimensional patch embeddings $v_p \in R^{T \times N \times D}$. A learnable [class] token is prepended to the patch embeddings $v_p$ as $v_0 \in R^{T \times (N+1) \times D}$. To retain position information, a learnable positional embedding $E_{pos} \in R^{(N+1) \times D}$ is added to $v_0$ as $z_0 = v_0 + E_{pos}$, where the $E_{pos}$ is broadcast to $T$ dimension to be shared among different frames and $z_0$ is the final input of the stacked ViT blocks. Each ViT block consists of LayerNorm (LN), multi-head self-attention (MSA) and MLP layers, with residual connections after each block, as shown in Figure 2 (b). The computation process during each ViT block can be given by:

$$z'_l = \text{MSA}(\text{LN}(z_{l-1})) + z_{l-1} \qquad (1)$$

$$z_l = \text{MLP}(\text{LN}(z'_l)) + z'_l \qquad (2)$$

where $z_l$ denotes the output of the $l$-th ViT block and all the MSAs are computed on patch dimension to learn the relationship of different spatial parts of the original frame. The [class] tokens of each frame from the last ViT block are concatenated as the final video feature $F_v \in R^{T \times D}$ which is fed into the classifier and used to calculate the final classification score.

Inspired by PEFT techniques, AIM [54] tries to adapt pre-trained image model to video action recognition with minimal fine-tuning by freezing the original model and only fine-tuning additionally introduced Adapters shown in Figure 2 (a). Moreover, AIM proposes to reuse the pre-trained MSA layer to do temporal modeling which is deficient in the vanilla ViT. As shown in Figure 2 (c), a shared frozen temporal MSA (T-MSA) layer is prepended to the original spatial MSA (S-MSA) sequentially to exchange the temporal information between frames for temporal modeling, which can be achieved easily by permuting the patch dimension and the

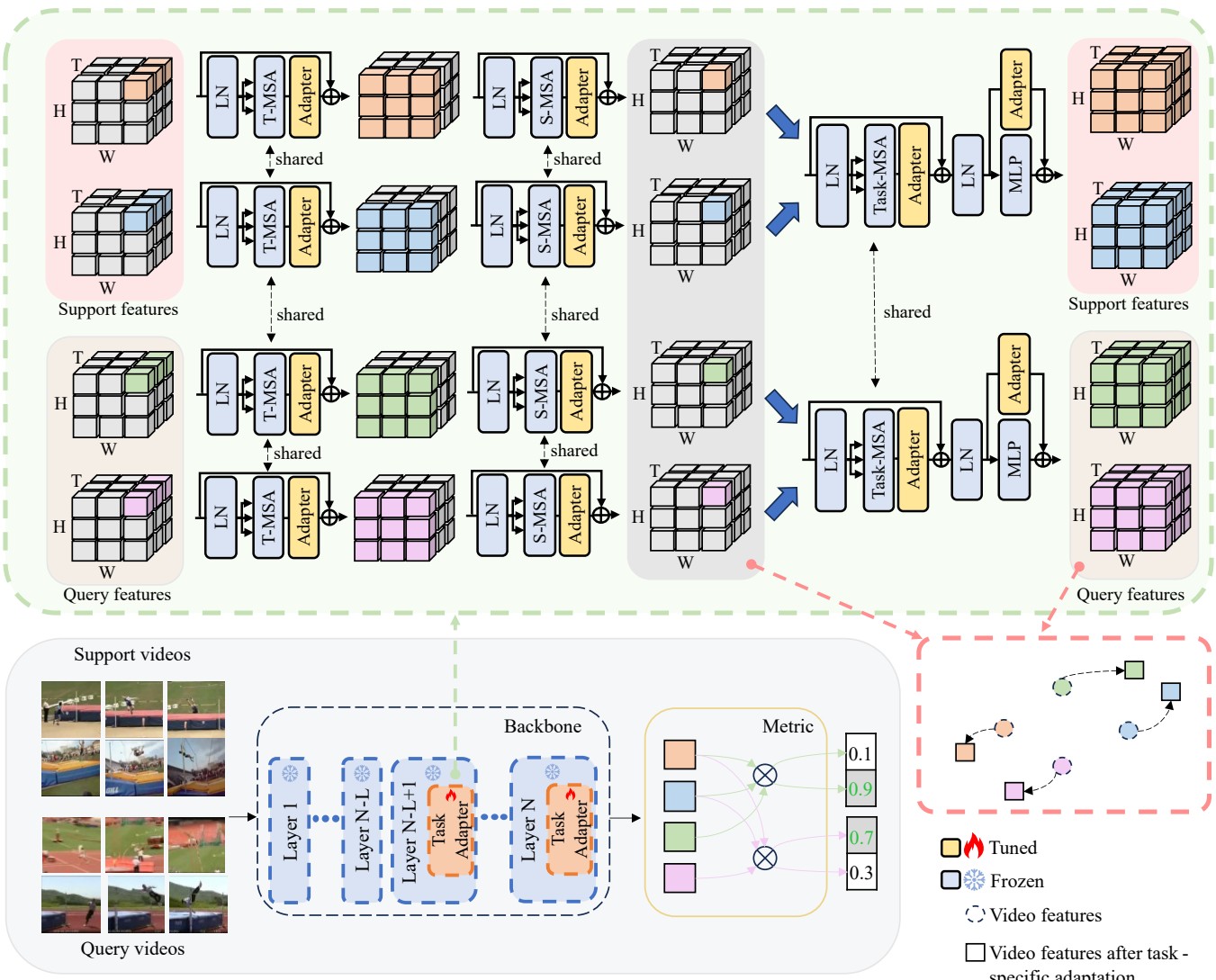

**Figure 3: Illustration of our method. Note that we only add Adapters into the last $L$ ViT layers. In Task-Adapter, we introduce task adaptation after T-MSA and S-MSA to enhance the task-specific information for the few-shot action recognition. After feature extraction, the video features are passed to the metric module to compute the classification scores. The upper figure illustrates the computational process of the Task-Adapter given a 2-way 1-shot learning task with two query videos in the query set.**

temporal dimension. The tunable Adapter is added after each MSA layer and parallel to the MLP layer to achieve temporal, spatial and joint adaptation as follows:

$$z_l^t = \text{Adapter}(\text{T-MSA}(\text{LN}(z_{l-1}))) + z_{l-1} \tag{3}$$

$$z_l^s = \text{Adapter}(\text{S-MSA}(\text{LN}(z_l^t))) + z_l^t \tag{4}$$

$$z_l = \text{Adapter}((\text{LN}(z_l^s)) + \text{MLP}(\text{LN}(z_l^s)) + z_l^s \tag{5}$$

where $z_l^t$, $z_l^s$ and $z_l$ individually stands for temporal adapted, spatial adapted and jointly adapted output from the $l$-th ViT block. However, directly transferring this method to few-shot action recognition is inadequate due to the lack of considering the relationship

between different videos within the few-shot learning task and the information gap between seen classes and unseen classes.

## 3.2 Adapting ViT to Task-specific FSAR

Few-shot action recognition (FSAR) studies how to classify unseen class query videos with only a limited number of labeled samples available. Following the episodic training [39], we randomly sample a $C$-way $K$-shot task, which has $C$ action classes and $K$ labeled samples in each class, from the training set $\mathcal{D}_{tr}$ for each episode during training. All the $C \times K$ video samples form the support set $\mathcal{S} = \{x_i\}_{i=1}^{C \times K}$. Additionally, there is a query set $\mathcal{Q} = \{x_i\}_{i=1}^{Q}$ from the same $C$ action classes. The learning objective of each task is to

classify each query video $q \in Q$ into one of the $C$ classes. Although the temporal adaptation proposed by AIM gives vanilla ViT the ability of temporal modeling, it remains unaddressed that the ViT model could not attend to different videos and enhance the task-specific discriminative features.

Next, we describe how we empower the frozen ViT with the ability to adapt embeddings with task-specific information for few-shot action recognition. As described in Section 3.1, we first map all the videos in the whole sampled task (both support set and query set) into their patch embeddings by Eq. 1 and Eq. 2, then we get the input features of the first ViT block $\mathcal{S}_0 = [x_0^1, ..., x_0^{CK}]$ and $Q_0 = [x_0^1, ..., x_0^Q]$, where $x_0^i \in R^{T \times (N+1) \times D}$ denotes the patch embeddings of the $i$-th video both for the support set and query set. Then, these patch embeddings are fed into several frozen ViT blocks to exchange spatial information between different tokens. It should be noted that we only introduce tunable adapters in the last $L$ ViT blocks, which is a choice based on empirical study. We will further analyze the impact of this decision in the ablation experiment. Starting from the $(N-L+1)$-th layer, we introduce tunable adapter layers into the original model. Specifically, besides T-MSA and S-MSA, we further reuse the frozen MSA layer as Task-MSA after the temporal and spatial adaptation to perform task-specific self-attention across the tokens at the same spatial-temporal location in different videos (see Figure 3, upper part). Take the support features as an example:

$$\mathcal{S}_l^t = \text{Adapter}(\text{T-MSA}(\text{LN}(\mathcal{S}_{l-1}))) + \mathcal{S}_{l-1} \quad (6)$$

$$\mathcal{S}_l^s = \text{Adapter}(\text{S-MSA}(\text{LN}(\mathcal{S}_l^t))) + \mathcal{S}_l^t \quad (7)$$

$$\mathcal{S}_l^{task} = \text{Adapter}(\text{Task-MSA}(\text{LN}(\mathcal{S}_l^s))) + \mathcal{S}_l^s \quad (8)$$

$$\mathcal{S}_l = \text{Adapter}(\text{LN}(\mathcal{S}_l^{task})) + \text{MLP}(\text{LN}(\mathcal{S}_l^{task})) + \mathcal{S}_l^{task} \quad (9)$$

where $S_{l-1}, S_l \in R^{CK \times T \times (N+1) \times D}$ individually denotes the input and output support features of the $l$-th layer ($l \in \{N - L + 1, N - L + 2, ..., N\}$).

The query features can be computed in the same way. To avoid exposing the target support class features to the specific query video feature, we isolate the information interaction between the support set and query set. The bottom right corner of Figure 3 shows a conceptual view illustrating the changes that task-specific self-attention brings to the video features. As can be seen, the support video features of different classes are pushed away, and the query video features are pulled closer to the video features of the target support class based on the unique information specific to each task, resulting in varying effects across different tasks. As a result, we explicitly enhance the the most discriminative features in the given few-shot action recognition task with the introduced task-specific adaptation.

Finally, we concatenate the [class] tokens of each frame from the last ViT block to get the support set video representations $\mathcal{F}_\mathcal{S} \in R^{C \times K \times T \times D}$ and the query set video representations $\mathcal{F}_Q \in R^{Q \times T \times D}$. The final question is how to compute the distance between the query features $\mathcal{F}_Q$ and each class using the support features $\mathcal{F}_\mathcal{S}$. The existing works further design elaborate temporal alignment modules to construct class prototypes for each class, aiming to align the action semantics at the feature level. Thanks to our proposed task-specific adaptation, we have completed the semantic alignment

during the process of feature extraction. Consequently, we directly average all the support video features as the class prototype for each class:

$$\hat{\mathcal{F}}_\mathcal{S}^c = \frac{1}{K} \sum_i^K \mathcal{F}_\mathcal{S}[c][i] \quad (10)$$

where $\hat{\mathcal{F}}_\mathcal{S}^c \in R^{C \times T \times D}$ denotes the class prototype of the $c$-th support class.

To calculate the classification probability, we measure the distance between the query features and the class prototypes by an arbitrary metric module, e.g., TRX [31], OTAM [4] and Bi-MHM [44], which can formulated by:

$$\mathcal{P}_i^c = \text{Metric}(\mathcal{F}_Q[i], \hat{\mathcal{F}}_\mathcal{S}) \quad (11)$$

where $\mathcal{P}_i^c$ is the probability of the $c$-th class for the $i$-th query video in query set and $\hat{\mathcal{F}}_\mathcal{S} = [\hat{\mathcal{F}}_\mathcal{S}^1, \hat{\mathcal{F}}_\mathcal{S}^2, ..., \hat{\mathcal{F}}_\mathcal{S}^c] \in R^{C \times T \times D}$ stands for all the class prototypes of the whole support set.

For the metric modules, we will demonstrate in our experiments that our proposed Task-Adapter consistently improves the performance of any metric methods. Besides that, owing to the superiority of our task-specific adaptation during feature extraction, excellent performance can be achieved simply by averaging the frame-to-frame cosine similarity without complex metric measurements. During the training stage, classification probabilities for each task serve as the logits for computing cross-entropy loss and backpropagating gradients. Importantly, we solely fine-tune the parameters of the additional adapter layers while keeping the original pre-trained model frozen. In testing, we freeze the entire backbone, including the adapters, and directly extract features for the unseen test classes.

## 4 EXPERIMENTS

### 4.1 Experimental Setup

**Datasets.** We evaluate our method on four datasets that widely used in few-shot action recognition, namely HMDB51 [16], UCF101 [35], Kinetics [5] and SSv2 [11]. The first three datasets focus on scene understanding, while the last dataset, SSv2, has been shown to require more challenging temporal modeling [4, 38]. Particularly, We use the few-shot split proposed by [57] for HMDB51 and UCF101, with 31/10/10 classes and 70/10/21 classes for train/val/test, respectively. Following [60], Kinetics is used by selecting a subset which consists of 64, 12 and 24 training, validation and testing classes. For SSv2, we provide the results of two publicly available few-shot split, i.e., SSv2-Small [61] and SSv2-Full[4]. Both SSv2-Small and SSv2-Full contain 100 classes selected from the original dataset, with 64/12/24 classes for train/val/test, while SSv2-Full contains 10x more videos per class in the training set.

**Implementation details.** As previously illustrated, we aim to adapt the image pre-trained ViT model to achieve task-specific few-shot action recognition. To show the applicability of our method for different pre-trained models, we respectively choose the ImageNet pre-trained [7] and CLIP pre-trained [32] ViT as our backbone. Specifically, for CLIP pre-trained backbone, we combine the zero-shot results and few-shot results in the same way as CLIP-FSAR [42]

Table 1: Comparison with current SOTA few-shot action recognition methods on 5-way 1-shot and 5-way 5-shot benchmarks. The reported results cover both temporal-related dataset (SSv2) and scene-related datasets (including HMDB51, UCF101, and Kinetics). The best results are highlighted in bold and the second-best results are underlined.

| Method | Reference | Backbone | Pretrain | Fine-tuning | SSv2-Small | | SSv2-Full | | HMDB51 | | UCF101 | | Kinetics | |
|---|---|---|---|---|---|---|---|---|---|---|---|---|---|---|
| | | | | | 1-shot | 5-shot | 1-shot | 5-shot | 1-shot | 5-shot | 1-shot | 5-shot | 1-shot | 5-shot |
| CMN [60] | ECCV'18 | ResNet50 | IN-21K | Full Fine-tuning | 34.4 | 43.8 | 36.2 | 48.9 | - | - | - | - | 60.5 | 78.9 |
| TARN [26] | BMVC'19 | C3D | Sports-1M | Full Fine-tuning | - | - | - | - | - | - | - | - | 66.6 | 80.7 |
| ARN [57] | ECCV'20 | C3D | - | Full Fine-tuning | - | - | - | - | 44.6 | 59.1 | 62.1 | 84.8 | 63.7 | 82.4 |
| OTAM [4] | CVPR'20 | ResNet50 | IN-21K | Full Fine-tuning | - | - | 42.8 | 52.3 | - | - | - | - | 73.0 | 85.8 |
| AmeFuNet [9] | MM'20 | ResNet50 | IN-21K | Full Fine-tuning | - | - | - | - | 60.2 | 75.5 | 85.1 | 95.5 | 74.1 | 86.8 |
| TRX [31] | CVPR'21 | ResNet50 | IN-21K | Full Fine-tuning | - | 59.1 | - | 64.6 | - | 75.6 | - | 96.1 | 63.6 | 85.9 |
| SPRN [41] | MM'21 | ResNet50 | IN-21K | Full Fine-tuning | - | - | - | - | 61.6 | 76.2 | 86.5 | 95.8 | 75.2 | 87.1 |
| HyRSM [44] | CVPR'22 | ResNet50 | IN-21K | Full Fine-tuning | 40.6 | 56.1 | 54.3 | 69.0 | 60.3 | 76.0 | 83.9 | 94.7 | 73.7 | 86.1 |
| TA$^2$N + Sampler [22] | MM'22 | ResNet50 | IN-21K | Full Fine-tuning | - | - | 47.1 | 61.6 | 59.9 | 73.5 | 83.5 | 96.0 | 73.6 | 86.2 |
| MoLo [43] | CVPR'23 | ResNet50 | IN-21K | Full Fine-tuning | 41.9 | 56.2 | 55.0 | 69.6 | 60.8 | 77.4 | 86.0 | 95.5 | 74.0 | 85.6 |
| MASTAF [23] | WACV'23 | ViViT | JFT | Full Fine-tuning | 45.6 | - | 60.7 | - | 69.5 | - | 91.6 | - | - | - |
| MGCSM [56] | MM'23 | ResNet50 | IN-21K | Full Fine-tuning | - | - | - | - | 61.3 | 79.3 | 86.5 | 97.1 | 74.2 | 88.2 |
| SA-CT [58] | MM'23 | ResNet50 | IN-21K | Full Fine-tuning | - | - | 48.9 | 69.1 | 60.4 | 78.3 | 85.4 | 96.4 | 71.9 | 87.1 |
| SA-CT(ViT) [58] | MM'23 | ViT-B | IN-21K | Full Fine-tuning | - | - | - | 66.3 | - | 81.6 | - | 98.0 | - | 91.2 |
| CLIP-FSAR [42] | IJCV'23 | ViT-B | CLIP | Full Fine-tuning | 54.6 | 61.8 | 62.1 | 72.1 | 75.8 | 87.7 | 96.6 | 99.0 | 89.7 | 95.0 |
| MA-CLIP [51] | ArXiv'23 | ViT-B | CLIP | PEFT | 59.1 | 64.5 | 63.3 | 72.3 | **83.4** | 87.9 | 96.5 | 98.6 | **95.7** | 96.0 |
| **Task-Adapter(Ours)** | - | ViT-B | CLIP | PEFT | **60.2** | **70.2** | **71.3** | **74.2** | 83.2 | **88.8** | **98.0** | **99.4** | 95.0 | **96.8** |

which uses a simple element-wise multiplication. For a fair comparison with the existing works [42, 51], we uniformly sample 8 frames for each video, and crop the frames to 224×224 as the input resolution. During the training stage, random crop is used for data augmentation. We freeze the original pre-trained model and only fine-tune the introduced adapters (as shown in Figure 2 (c)) using the SGD optimizer with a learning rate of 0.001. For the testing stage, we freeze all the parameters of our backbone to extract task-specific discriminative video features for measuring the distances. All reported results in our paper are the average accuracy over 10,000 few-shot action recognition tasks.

## 4.2 Comparison with state-of-the-art methods

Table 1 presents a comprehensive comparison of our method with recent few-shot action recognition methods. As our Task-Adapter is designed for ViT-based architectures, we use CLIP [32] pre-trained ViT to make a fair comparison with recent few-shot action recognition works that use large-scale pretrained models. For the scene-related datasets, the background understanding is more crucial. Of particular note, by comparing the results of works that use CLIP pre-trained models and those that use ImageNet (IN-21K) pre-trained models, we can conclude that the performance of few-shot action recognition indeed benefits from large-scale pretraining, especially for scene-related datasets. As shown in Table 1, our Task-Adapter can achieve promising results on HMDB51, UCF101 and Kinetics. Especially for 5-shot tasks, our method outperforms the exsisting SOTA methods by 0.9% on HMDB51, 0.4% on UCF101 and 0.8% on Kinetics, respectively. Compared with full fine-tuning method CLIP-FSAR [42], we can see our method has an overall improvement (from 0.4% to 7.4%) on all datasets. Compared with PEFT method MA-CLIP [51], our method has an improvement of 0.9%, 0.8% and 0.8% respectively on HMDB51, UCF101 and Kinetics in 5-shot setting. Note that while our method exhibits slightly lower 1-shot performance on HMDB51 and Kinetics compared to MA-CLIP (0.2% and 0.7% respectively), it achieves a 1.5% improvement

on UCF101 in the 1-shot setting. We attribute this to our under-utilization of the semantic information in HMDB51 and Kinetics, while MA-CLIP design an elaborate multimodal adaptation method with text-guided prototype construction module. We would like to consider more comprehensive multi-modal fusion as the future research.

As mentioned in Section 4.1, SSv2 requires more temporal modeling than other three datasets, making it much more challenging. In our experiments, we observe that our proposed task-specific adaptation significantly enhances the temporal alignment during the feature extraction process. From Table 1, we can see that our method significantly outperforms all existing approaches on both SSv2-small and SSv2-Full, establishing new state-of-the-art results. Except for MA-CLIP [51] and ours, most of the previous works fully fine-tune the pre-trained models, resulting in inferior results due to the overfitting problem. Most relevant to our method, MA-CLIP [51] is also based on AIM [54], but it focuses on the multimodal adaptation mechanism at the feature level, while our method emphasizes the importance of task-specific adaptation during the feature extraction process. The results on the SSv2 datasets demonstrate the effectiveness of our proposed method, showing a significant improvement of 5.7% for the 5-shot task on SSv2-Small and 8.0% for the 1-shot task on SSv2-Full, respectively. Also, we can see that the improvement of the 1-shot task on SSv2-Small is smaller than that on SSv2-Full. We attribute this phenomenon to the 10x increase in the number of videos for each class in SSv2-Full, which enables our Task-Adapter to learn low-shot task-specific adaptation through a variety of sampled tasks.

## 4.3 Applicability of proposed Task-Adapter

In Table 2, we demonstrate the applicability of our method on different pre-trained weights. From the upper part of the table, we show the performance of different fine-tuning methods with IN-21K pre-trained weights. Note that the supervised pretraining on ImageNet does not follow the contrastive text-image pretraining

**Table 2: Illustration of the effectiveness of our Task-Adapter using different pretrained weights (e.g., IN-21K pre-trained and CLIP pre-trained weights). The experiment is conducted on 5-way 1-shot task on UCF101.**

| Methods | Bacbone | Pretrain | Modality | Acc |
|---|---|---|---|---|
| Full Fine-tuning | ViT-B | IN-21K | Unimodal | 73.0 |
| AIM Adapter | ViT-B | IN-21K | Unimodal | 84.7 |
| Task Adapter | ViT-B | IN-21K | Unimodal | 87.7 |
| Full Fine-tuning | ViT-B | CLIP | Multimodal | 93.5 |
| AIM Adapter | ViT-B | CLIP | Multimodal | 96.5 |
| Task Adapter | ViT-B | CLIP | Multimodal | 98.0 |

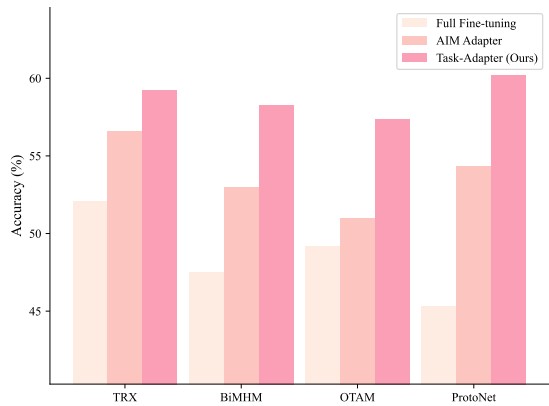

**Figure 4: Comparison of the performance achieved by combining different fine-tuning strategies with existing widely used metric measurement methods for 5-way 1-shot task on the challenging SSv2-Small.**

paradigm used by CLIP. Consequently, we can only perform unimodal inference with pure visual information. As can be seen, just substituting the full fine-tuning with the PEFT method of AIM brings an improvement of 9.7%, which is in line with our insights that full fine-tuning limits the generalization capability of the pretrained model. Furthermore, an another 3.0% improvement (totally up to 10.7%) is observed when using our Task-Adapter due to the fact that it enhances the capability of extracting task-specific information from the few-shot learning task. Also, we can see a 1.5% improvement of our Task Adapter over AIM and a 4.5% improvement over full fine-tuning under CLIP pre-trained weights. The consistent improvement over both full fine-tuning and AIM under different pre-trained weights show the strong generalization capability of our method.

We also compare the performance of different fine-tuning strategies using different metric measurement methods to demonstrate the applicability of our Task-Adapter. Four commonly used metric measurement methods are taken in this paper, namely TRX [31], Bi-MHM [44], OTAM [4] and ProtoNet (e.g., directly averaging the frame-to-frame cosine similarity over temporal dimension without complex metric measurements). Taking full fine-tuning strategy as the baseline, we observe a consistent improvement across all metric measurement methods, as shown in Figure 4. It should be noted

**Table 3: Effectiveness of each component. We choose frozen backbone as the baseline and compare the performance of different settings for 5-way 1-shot task on SSv2-Small.**

| Frozen Backbone | AIM | Task-specific Adaptation | Partial Adapting | Acc |
|---|---|---|---|---|
| ✓ | ✗ | ✗ | ✗ | 36.0 |
| ✗ | ✗ | ✗ | ✗ | 47.8 |
| ✓ | ✓ | ✗ | ✗ | 53.3 |
| ✓ | ✓ | ✗ | ✓ | 54.3 |
| ✓ | ✓ | ✓ | ✗ | 55.2 |
| ✓ | ✗ | ✓ | ✓ | 56.6 |
| ✓ | ✓ | ✓ | ✓ | **60.2** |

**Table 4: Effect of inserting position of Task-Adapters on SSv2-Small.**

| Position | Tunable Param | Acc |
|---|---|---|
| All | 14.1 M | 55.2 |
| Bottom 6 | 7.2 M | 50.7 |
| Top 6 | 7.2 M | 60.2 |

**Table 5: Impact of position of Task-MSA relative to T&S-MSA on SSv2-Small.**

| Methods | Acc |
|---|---|
| in front of T&S-MSA | 55.6 |
| between T&S-MSA | 55.5 |
| back of T&S-MSA | 60.2 |

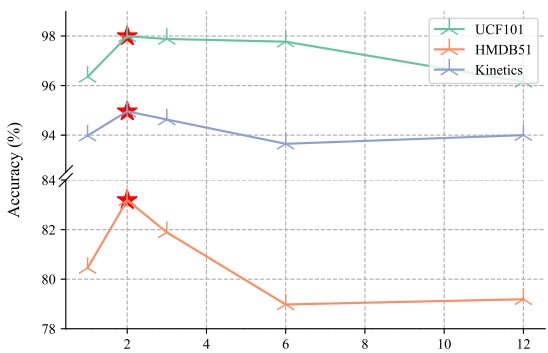

**Figure 5: Effect of inserting Task-Adapters into the last $L$ ViT layers (e.g., $L$ = 1, 2, 3, 6, 12) on scene-related datasets.**

that ProtoNet outperforms all the other methods when uniformly using Task-Adapter. We attribute it to the fact that with our Task-Adapter, the model is able to dynamically align the task-specific feature during the process of feature extraction corresponding to the given task. Therefore it is unnecessary to add extra complex alignment method at feature level.

## 4.4 Ablation Studies

**Effectiveness of each component.** To demonstrate the effectiveness of each component, we perform detailed ablation experiments in Table 3. The first row shows the results of a frozen pre-trained model without any learnable adapters, while the second row refers to the performance of full fine-tuning. Comparing the results of the third and fourth rows, we can see the effectiveness of PEFT over traditional full fine-tuning in few-shot action recognition. Note that the "Partial Adapting" in the fourth column of Table 3 refers to a common strategy which only adds adapters in the last several ViT layers. We can observe from the table that the improvement of introducing task-specific adaptation into AIM is distinct (1.9%) when inserting adapters into all the layers of the pre-trained model. Additionally, using partial adapting strategy for AIM also brings

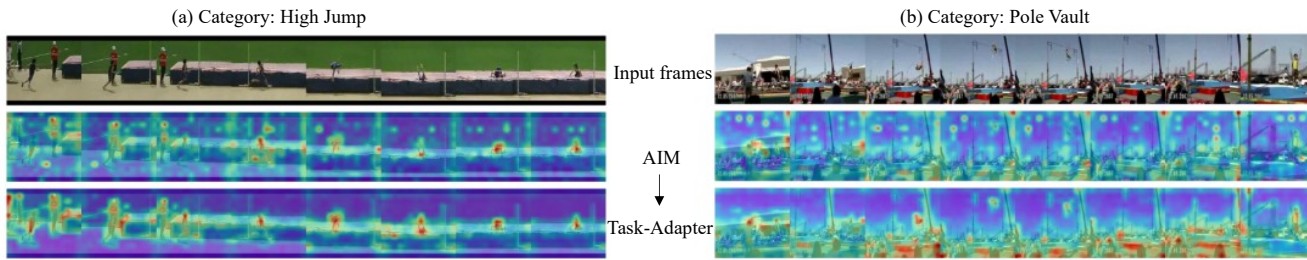

**Figure 6: Visualizations of the attention map of "High Jump" and "Pole Vault" for a given few-shot learning task obtained by baseline AIM and our Task-Adapter. Our method is able to pay more attention to the most discriminative area of the actions with the help of task-specific adaptation.**

an improvement of 1%. Even so, the improvement of introducing task-specific adaptation into AIM is notably enhanced under the partial adapting strategy, increasing from 1.9% to 5.9%. As mentioned in Section 3.1, AIM brings temporal modeling ability to the pre-trained image model, while our Task-Adapter further equip the model with the capability of performing task-specific adaptation. The result of the second-to-last row shows that we can still achieve comparable results only using task-specific adaptation without any temporal modeling, which demonstrates the importance of finding task-specific information for the few-shot learning task. Finally, the complete version of our method which only adds Adapters into the last several layers to the frozen model achieves the best performance.

**Partial Adapting.** In this section, we conduct ablation studies on the partial adapting strategy. Firstly, we study the effect of different inserting positions of our Task-Adapter in Table 4. The default setting is inserting adapters into all of the 12 ViT layers. Alternatively, we also evaluate the settings which respectively insert adapters to the bottom 6 layers and top 6 layers of the model (assuming forward propagation is from bottom to top). Inserting adapters only into the top 6 layers has been shown to achieve comparable performance with inserting to all layers in AIM [54]. However, inserting our Task-Adapter only into the top 6 layers outperforms inserting it into all layers by a large margin (5%) for few-shot action recognition. We attribute this to the hypothesize that bottom layers focus on generic features that do not require extensive adaptation, while the top layers focus on task-specific and discriminative features that benefit a lot from our task-specific adaptation module. Note that even with our Task-adapter inserted into all the layers, the increase in tunable parameters is small (only 14 M compared to 400 M frozen parameters of the pretrained model), which demonstrates the training efficiency of our method. For the scene-related dataset, we choose the best partial adapting strategy by ablation experiments depicted in Figure 5. We can see from the figure that inserting our Task-Adapter into the top 2 layers is the best setting for all these three datasets. Since these scene-related datasets focus on background understanding and do not need much temporal modeling, a more lightweight setting with inserting adapters into only 2 top layers is more appropriate.

**Position of Task-MSA relative to AIM.** As illustrated in Section 3.1, the original AIM comprises two multi-head self-attentions (MSAs) individually for spatial modeling and temporal modeling. In this section, we study the optimal placement relative to the existing two MSAs of our Task-MSA for task-specific modeling. As

can be seen in Table 5, inserting the proposed Task-MSA after the T-MSA and S-MSA achieve the best performance. It should be noted that regardless of where the Task-MSA is inserted, it consistently outperforms the baseline AIM [54]. The results indicate that compared with AIM, the introduction of Task-MSA indeed improves the performance for few-short action recognition. Furthermore, the Task-MSA after T-MSA and S-MSA can maximize the ability to extract task-specific discriminative features within the task.

## 4.5 Visualizations

Figure 6 presents the attention map visualizations for a given few-shot learning task obtained by baseline AIM and our Task-Adapter. We visualize the action "High Jump" and "Pole Vault" from UCF101 in Figure 6 (a) and (b) respectively. As can be seen in Figure 6 (a), the attention maps of AIM attend to several background areas that are unassociated with the action being performed. After introducing the task-specific adaptation proposed by us, the model begins to focus on the movements of the actor which are the most discriminative features for the category of "High Jump". In Figure 6 (b), compared with the attention maps obtained by AIM, we can see that the areas of distinctive venue facilities gain a higher attention score (brighter color in the figure) with our Task-Adapter. For similar actions "High Jump" and "Pole Vault", as can be seen, the model begins to attend to the specific information of the actions, e.g, the movements for "High Jump" and the long pole and large foam mats for "Pole Vault" when task-specific information is considered. The visualization results demonstrate that our method can enhance the most discriminative features for the few-short action recognition.

## 5 CONCLUSION

In this paper, we propose a novel adaptation method named Task-Adapter to better adapt the pre-trained image models (including unimodal and multimodal pre-trained models) for few-shot action recognition. Instead of using the traditional full fine-tuning strategy, we propose to only fine-tune the newly inserted adapters which alleviates the issues of catastrophic forgetting and overfitting. To enhance the most discriminative features within the given few-shot learning tasks, we further propose to reuse the frozen self-attention layer for task-specific adaptation during the process of feature extraction, which remarkably improves the performance on four few-shot action recognition benchmarks and achieves the new state-of-the-art results, particularly on the challenging SSv2 dataset.

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
