# OpenReview forum: "Task-Adapter: Task-specific Adaptation of Image Models for Few-shot Action Recognition"
_acmmm.org/ACMMM/2024/Conference — MM2024 Poster_

### Official Review · Reviewer_uU8R · 2024-05-20

**Rating:** 4
**Confidence:** 3

**Summary:**

This work proposes a task-specific adaptation method (Task-Adapter) for few-shot action recognition. The key idea of the task-adapter method is to introduce task-specific adapters into the last several layers of the backbone and keeping the parameters of the original pre-trained model frozen. Experimental results on several standard FSL datasets are provided to evaluate the method.

**Strengths:**

+ The paper is well-written and easy to understand
+ The figures tables are well-designed.
+ Both qualitative and quantitative results are provided.

**Limitations:**

- The motivation is not clear. The proposed method are the choices it uses at/for different steps/components. These seem adhoc and not well motivated.
- More evidence should be provided (or at least discussed) to support the claim: "the overfitting problem can be resolved with the developed adapters". In my opinion, less trainable parameters may also result in the overfitting issue.
- The computational complexity and well as the limitations of the proposed method should be further discussed.
- It is interesting to further explore why the devised method achieves the best performance when L=2.

**Suitability:**

2

---

### Official Review · Reviewer_pfY4 · 2024-05-21

**Rating:** 3
**Confidence:** 3

**Summary:**

This paper proposes a method named Task-Adapter for few-shot action recognition. In contrast to AIM, the authors introduce Task-MSA between S-MSA and MLP. The experiments demonstrate that the Task-Adapter method is effective.

**Strengths:**

* Comaperd with SOTA methods, this method have a significant performance imporvement.

* The Ablation Studies prove the effectiveness of Task-Adapter.

* This paper is well-writen.

**Limitations:**

* The author claim that the video feature of different classes are pushed away and the same class is push closed. Could you please add visualization of the support and query fearture extracted by AIM and Task-adapter to better illustrate these dynamics.

* How Task-adater works when the number of classes in each few-shot task is 1. In such scenarios, there is no additional information for Task-Adapter to use to differentiate between the same and different classes.

* As mentioned aboved, I hope the author can add an ablation studies whose input to the first ViT block of Task-adapter is $x_0^1, \cdots, x_0^{CK} $ rather than $ S_0 = [x_0^1, \cdots, x_0^{CK}] $ for 5-shot setting. This experiment can prove the effectiveness of Task-adapter stems from task-specific information rather than merely from an increase in parameters.

**Suitability:**

1

---

### Official Review · Reviewer_12Fe · 2024-05-24

**Rating:** 4
**Confidence:** 3

**Summary:**

This paper proposes a simple yet effective task-specific adaptation method (Task-Adapter) for few-shot action recognition. Authors introduce the Task-Adapter into the last several layers of the backbone to perform task-specific self-attention across different videos.

**Strengths:**

The proposed method is parameter-efficient and effective
It performs task-specific adaptation during the process of feature extraction, which is different from the previous works.

**Limitations:**

The method may not be universally applicable. If the backbone is replaced by resnet-50 and the weights of CLIP are not used, is it still applicable and better than the previous FSAR method?

Authors inserted adapters into the top 6 layers, but did not prove that the task-specific adaptation is useful in all six layers. In other words, I think using AIM in top 6 layers and Task-MSA only in the last layer will have a good effect. In this case, line 179-182 are not convincing.

The results of the related work mentioned in line 250 should be compared in Table 1, since your method also involves multimodality (line 578-579). And the result of CLIP-FSAR in Table 1 should also be the ensemble one that combines the zero-shot and few-shot results.

**Suitability:**

3

---

### Official Review · Reviewer_vphx · 2024-05-25

**Rating:** 4
**Confidence:** 2

**Summary:**

The paper introduces a task-specific adaptation method, Task-Adapter, for few-shot action recognition, addressing overfitting issues associated with fully fine-tuning pre-trained models. By incorporating Task-Adapter into the backbone layers while keeping the original model parameters frozen, it enhances task-specific adaptation during feature extraction. Experimental results across four standard datasets, including SSv2, demonstrate significant performance improvements over state-of-the-art methods.

**Strengths:**

- **Efficient finetuning:** the efficient adaptation of large pretrained models is an interesting and relevant topic, and the paper proposes to study this in the challenging video action recognition task.
- **Experiments:** the experimental validation across several datasets, including the challenging SSv2, proves the effectiveness of the proposed method.

**Limitations:**

- **PEFT approaches:** the authors state that "the aforementioned PEFT methods do not consider the characteristic of the few-shot learning task, where task-specific adaptation plays a crucial role in distinguishing the query sample from irrelevant categories." (L270-273). It is not clear how the proposed adapter tackles and solves this problem and where other PEFT approaches fail at this.
- **Comparisons:** related to the previous point, it is not stated how other efficient CLIP adaptions for action recognition are not included in the quantitative comparison in Table 1, e.g. Vita-CLIP or AIM.

**Suitability:**

2

---

### Meta-Review · Area_Chair_kLjV · 2024-07-05

**Recommendation:** Accept (Poster)
**Confidence:** 5

**Metareview:**

This work introduces a task-specific adapter based model for few-shot action recognition. The proposed model can address overfitting issues associated with fully fine-tuning pre-trained models, and advance the alignment module into the process of feature extraction. This work is solid, and provides valuable insights regarding task-specific adaptation method (Task-Adapter) for few-shot action recognition. All the reviewers are satisfied with the author rebuttal. We encourage the authors to incorporate the reviewers' feedbacks into the revised version including computational complexity, limitations, additional ablation studies and analysis required by the reviewers.